# Modern Management and Diagnostics in HER2+ Breast Cancer with CNS Metastasis

**DOI:** 10.3390/cancers15112908

**Published:** 2023-05-25

**Authors:** Surbhi Warrior, Adam Cohen-Nowak, Priya Kumthekar

**Affiliations:** 1Department of Hematology, Oncology Northwestern Memorial Hospital, Chicago, IL 60611, USA; 2Department of Internal Medicine, Northwestern Memorial Hospital, Chicago, IL 60611, USA; 3Department of Neuro-Oncology, Northwestern Memorial Hospital, Chicago, IL 60611, USA

**Keywords:** breast cancer, HER2-positive, brain metastasis

## Abstract

**Simple Summary:**

HER2-positive breast cancer has an affinity for the central nervous system (CNS) manifested by metastases in both the brain parenchyma and the leptomeninges. The authors review the epidemiology of HER2-positive CNS metastases, risk factor, and prognosis herein. They also discuss available treatment options for these patients as well as treatments on the horizon.

**Abstract:**

Patients with HER2-positive breast cancer have seen improved survival and outcomes over the past two decades. As patients live longer, the incidence of CNS metastases has increased in this population. The authors’ review outlines the most current data in HER2-positive brain and leptomeningeal metastases and discuss the current treatment paradigm in this disease. Up to 55% of HER2-positive breast cancer patients go on to experience CNS metastases. They may present with a variety of focal neurologic symptoms, such as speech changes or weakness, and may also have more diffuse symptoms related to high intracranial pressure, such as headaches, nausea, or vomiting. Treatment can include focal treatments, such as surgical resection or radiation (focal or whole-brain radiation), as well as systemic therapy options or even intrathecal therapy in the case of leptomeningeal disease. There have been multiple advancements in systemic therapy for these patients over the past few years, including the availability of tucatinib and trastuzumab-deruxtecan. Hope remains high as clinical trials for CNS metastases receive greater attention and as other HER2-directed methods are being studied in clinical trials with the goal of better outcomes for these patients.

## 1. Epidemiology of Breast Cancer and CNS Metastasis

Significant improvement in systemic breast cancer therapies have led to increased survival rates. Despite these advances, the incidence and mortality of central nervous system (CNS) metastases continues to rise, likely due to poor penetration of these treatments through the blood–brain barrier [1]. Breast cancer is the second most common cause of brain metastases, with incidence ranging from 10–30% among all breast cancer patients [2,3]. Risk factors for progression to brain metastases include young age, age greater than 70, more than two areas of metastatic disease at diagnosis, larger tumor size at diagnosis, and hormone receptor negativity [4]. The most common subtypes of breast cancer that have the highest rates of CNS disease include triple negative, luminal type B, and Human Epidermal Growth Factor 2 (HER2)-positive tumors [5]. 

HER2 is overamplified in about 25% of all breast cancers [6]. The HER2 gene is a protein tyrosine kinase that causes dimerization, and when overexpressed promotes breast cancer cell survival through increased cellular signaling (Figure 1). HER2 positivity is analyzed via immunohistochemistry for expression of the HER2 protein (considered positive when IHC 3+ or greater) or by HER2 gene amplification by fluorescence in situ hybridization (FISH) gene copy measurement of 6 or more, or HER2/CEP17 ratio of 2 or higher. Data show that overexpression of HER2 can help provide prognostic guidance in breast cancer patients. Breast cancers that have amplification of HER2 can have worse overall survival and more aggressive disease [6]. The metastatic profile for HER2-positive breast cancer patients is more likely to include spread to the lung, liver, visceral organs, and brain [7]. Improvement in current therapies provide HER2 domain targetable treatment options for these patients and are continuously evolving. Systemic treatment for HER2-positive breast cancer patients can include tyrosine kinase inhibitors (TKIs), monoclonal antibodies, and antibody drug conjugates (ADCs), as well as combinations of chemotherapy. Understanding the sequence of treatment regimens as well as mechanisms of resistance while balancing toxicities can be challenging. 

In HER2-positive breast cancer, up to 55% of patients develop CNS metastases during their disease course [8]. Interestingly, when brain metastases are identified, systemic disease is either stable or responds to trastuzumab-based therapy in up to 50% of patients [9]. This has been better understood as HER2-targeted monoclonal antibodies, such as trastuzumab, have been approved and are extremely effective at treating systemic disease, but have low efficacy at treating brain metastases due to impaired ability to cross the blood–brain barrier [8]. Studies have shown that development of brain metastases has less to do with decreased HER2 overexpression in brain lesions, showing that they can be targeted with HER2-directed therapy that can penetrate the CNS [3]. 

There is still significant uncertainty on how cancer cells are able to proliferate from the breast parenchyma and embed into the CNS, forming brain metastases. The metabolic, immune, anatomic, and cellular environment of the brain is unique from other areas of the body, making cancer cells that are able to metastasize to the brain very unique and advanced [10]. Cancer cells require genetic evolution in order to be able to infiltrate and adapt to the brain’s microenvironment [10]. After hematologic spread from the breast tissue, cancer cells must be able to cross the blood–brain barrier, which is a challenging obstacle made of capillary endothelial cells, a neuroglial membrane, astrocytes, pericytes, and glial podocytes [11]. Prior treatment for breast cancer, such as radiotherapy, can also lead to disruption in the blood–brain barrier and allow cancer cells to cross with more ease. Upon reaching the CNS, the breast cancer cells are able to invade tissue in the brain via extravasation and have perivascular growth [12]. In order to survive in the brain, breast cancer metastases may develop properties that allow them to assume overexpression of proteins to adapt to the brain’s microenvironment [13]. Throughout tumor progression in the brain, there is also a component of angiogenesis which leads to disruption of the blood–brain barrier, leading to heterogeneity in treatment penetration [12].

This review will focus on the clinical presentation and diagnosis of brain metastases in HER2-positive breast cancer patients, a review of significant clinical trials that affect management decisions, and treatment with local and/or systemic therapy with anti-HER2-directed agents, as well as prognosis and future directions in this evolving area of treatment of this cohort of patients. 

## 2. Clinical Presentation of CNS Metastasis and Diagnostics

Diagnosing metastatic disease in the brain begins with assessing the patient’s symptoms as well as neuroimaging. Presenting symptoms usually include headaches, confusion, mental status, changes in behavior, or any focal neurologic changes. Signs specific to tumor location include seizures, motor deficits, changes in sensation, difficulty with speech, and visual disturbances. Less commonly, patients also have lesions in the brain stem and cerebellum which have symptoms such as ataxia, upper-motor-neuron-related changes, cranial-nerve-related neuropathies, and hydrocephalus. Presence of cerebral edema due to tumor burden can cause severe nausea and vomiting as well. 

Imaging to further characterize brain lesions requires contrast-enhanced imaging. This can be in the form of computed tomography (CT) scans or magnetic resonance imaging (MRI). Further work-up to understand the presence of cancer cells in the cerebrospinal fluid cytology with a lumbar puncture or positron emission tomography (PET) scan can be performed if there are more global symptoms. Changes in mental status or cognition, headaches, or neuropathies can raise concern for leptomeningeal disease and indicate cerebrospinal fluid/subarachnoid space involvement [14]. 

Immediate ancillary neurologic care can be necessary in patients diagnosed with new breast cancer brain metastasis who have symptomatic disease. Evidence of cerebral edema seen on imaging in combination with brain-parenchyma-associated tumors, such as headaches, nausea, and vomiting, should lead to consideration of treatment with corticosteroids. Studies have shown that treatment with steroids can significantly reduce peritumoral edema within a few days. Anderson et al. reported a decrease in total edema in the brain of up to 10.3% after 7 days of treatment with steroids, and the effect of steroids was observed up to 63 days after treatment [15]. Dexamethasone is recommended as the steroid of choice for symptomatic relief in brain metastases [16]. The American Society of Clinical Oncology (ASCO) recommends treatment with dexamethasone 4 to 8 mg/d to decrease intracranial pressure and provide symptom relief related to tumor mass effect [17,18]. Doses as high as 16 mg/d of dexamethasone should be considered if there are severe symptoms related to increased intracranial pressure [17,18]. While steroids can provide immediate relief of symptoms, they should also be balanced with tapering as soon as clinically feasible and tolerated to prevent long-term side effects of steroids, such as, but not limited to, adrenal insufficiency, elevated blood sugars, and osteoporosis, amongst other steroid toxicities [19]. Steroids should also be avoided at night due to difficulty with agitation and insomnia. It is not recommended to treat patients who have metastatic brain metastasis with steroids who are asymptomatic and show no signs of mass effect [17].

Treatment with prophylactic antiepileptic drugs was initially considered for patients with breast-cancer-related brain metastases. Data have shown no survival benefits or decrease in seizure prophylaxis with the use of antiepileptic drugs in this patient population [20]. A randomized control trial by Forsyth et al. evaluated patients with brain tumors, no prior seizures, and who did not undergo surgery, receiving antiepileptic drugs or not, and the results showed that seizures occurred in 26% of patients treated with antiepileptics compared to 15% in the non-antiepileptic group [21]. A study by Wu et al. examined randomized patients with brain metastases who were undergoing surgery to have perioperative seizure prophylaxis treated with antiepileptic drugs versus not, and the results showed that 24% of patients in the antiepileptic drug group suffered seizures versus 18% of patients who had seizures in the non-treatment group [22]. As such, it is not recommended to employ the routine use of prophylactic antiepileptic drugs in patients with brain metastasis who have not previously had seizures and who are not perioperatively given antiepileptics for craniotomy [20].

## 3. Treatment of HER2+ Breast Cancer with CNS Metastasis

Treatment of HER2-positive brain metastases requires a multimodal approach to therapy based on overall prognosis, the state of current systemic disease, a patient’s functional status, the stage of brain disease, and location of the brain metastasis. Control of CNS disease from HER2-positive breast cancer includes use of surgery, radiation in the form of stereotactic radiosurgery (SRS) and whole-brain radiation therapy (WBRT), and systemic therapies that include HER2-targeted treatments. Advancements in HER2-targeted therapies have led to many options for treatment of this patient population. The sequence of treatment and which combination is appropriate for patients should be an individualized approach ideally decided on by a multidisciplinary team.

### 3.1. Local Therapy

Upfront decision on the treatment of local disease in the brain is made based on the extent of tumor burden in the brain, such as location, size, and number of lesions present. It is also necessary to assess a patient’s overall clinical picture with prognosis, performance status, and presenting symptoms. Surgery is recommended for patients with no other systemic disease or stable extracranial disease. It is also more commonly used when there is a solitary brain lesion that is causing peritumoral edema or any symptomatic mass effect. After surgery, a course of radiation should be performed to provide local control to the affected area [23]. 

SRS can also be used for treatment of solitary brain lesions when lesions are difficult to access for surgery or if there are multiple smaller lesions (typically for lesions 3 cm or smaller). SRS is also favored for treatment with patients with multiple but limited brain metastases. Previously, SRS was limited to patients who had 2–4 brain lesions, but new studies show that there is no difference in overall survival with patients with 5–10 brain metastases treated with SRS [23,24]. 

WBRT was previously the gold-standard treatment for patients with brain metastases, but more data show that it may have lower efficacy for local tumor control [25]. It is more effective for treatment of multifocal disease with greater than 5–10 lesions. WBRT can have significant side effects such as neurocognitive function impairment, transient increase in intracranial edema, hair loss, and overall impact on quality of life. There can be significant progressive short-term memory loss, typically occurring at least 6 months after treatment with WBRT, which can be irreversible [26]. This is thought to be due to exposure of the hippocampal region during radiation, and a more recent randomized study has shown that hippocampal-sparing WBRT with memantine may be a better option with less neurotoxicity [27,28].

### 3.2. Systemic Therapy

Systemic therapy is used to treat HER2-positive breast cancer patients with asymptomatic or limited brain metastases. It is also considered when there is progression of CNS disease after initial treatment with local therapies. There are multiple HER2-targeted therapies for metastatic breast cancer patients (Table 1). Patients with progressive brain metastases have generally been excluded from most clinical trials that have investigated HER2-targeted systemic therapies [29]. Commonly used HER2-targeted therapies, including monoclonal antibodies such as trastuzumab and pertuzumab, and antibody drug conjugates such as trastuzumab emtansine (T-DM1), have limited penetration across intact blood–brain barriers, making them less efficacious for treatment of CNS disease compared to their systemic efficacy [30]. 

Tyrosine kinase inhibitors (TKI) are smaller molecule HER2-targeted therapies that are able to penetrate the blood–brain barrier. Tucatinib is a TKI that was studied in the HER2CLIMB randomized control trial in combination with capecitabine and trastuzumab that showed progression-free survival, overall survival, and improved CNS response rates [31]. Data showed that one-year progression-free survival in the tucatinib arm was 24.9% compared to 0% in the placebo arm, the median duration of intracranial response was 6.8 months compared to 3.0 months, and median overall survival of 18.1 versus 12 months [31]. Because of the overt efficacy of tucatinib in the HER2CLIMB trial, further studies are being carried out to assess the response of tucatinib plus T-DM1 in patients with high-risk residual disease (NCT03975647). 

Trastuzumab deruxtecan (T-DXd) is an antibody–drug conjugate that showed strong efficacy in HER2-positive breast cancer patients that had previously received multiple lines of treatment [32]. The DESTINY-Breast01 trial showed superiority of treatment with T-DXd over T-DM1, with reduction of disease progression or death of 72% when compared with T-DM1, with reasonably tolerated side-effect profile [32]. The DEBBRAH trial evaluated the use of T-DXd in HER2-positive patients with brain metastasis or leptomeningeal carcinomatosis. The study showed that there was significant intracranial activity and response to treatment with tolerable toxicity and maintained quality of life in patients with new, stable, or progressive brain metastasis [33].

Even with limited penetration through the blood–brain barrier, the EMILIA trial showed that in treatment with T-DM1 in patients with brain metastasis, 16% of patients showed local progression compared to 22.2% in patients who were treated with lapatinib [34]. The overall survival was also significantly higher in patients treated with T-DM1 (HR, 0.38; 95% CI, 0.18–0.80; *p* = 0.008) [34]. The KAMILLA trial, a single-arm phase IIIb clinical trial, showed a 42% rate of intracranial response in HER2-positive breast cancer patients with brain metastasis [35]. 

Neratinib is another TKI that has shown to have significant CNS activity. The NALA trial studied Neratinib in combination with capecitabine compared to lapatinib, one of the initial TKIs used for targeting HER2-positive CNS disease, in combination with capecitabine. Results from the study showed improved progression-free survival in the neratinib arm (HR, 0.76; 95% CI, 0.63 to 0.93; *p* = 0.0059) [36]. Further evaluation with neratinib plus other targeted therapies are currently under investigation (NCT01494662). 

About 50% of HER2-positive breast cancer patients are also hormone-receptor-positive. Endocrine therapy is the mainstay of treatment for patients who are also estrogen- or progesterone-receptor-positive. Tamoxifen is a lipophilic agent that has some efficacy in penetrating the blood–brain barrier, but many times patients have hormone resistance when their disease progresses [37]. There is often discordance in the hormone receptor status between the initial tumor biology and in the sites of metastatic disease, with data showing discordance rates up to 40% [38]. Treatment with CDK4 and CDK6 inhibitors was thought to bypass hormonal resistance by targeting downstream pathways. Some data show that CDK4/6 inhibitors are able to cross the blood–brain barrier and can lead to clinical benefit of up to 25%; these agents are also under clinical trial in the setting of brain metastases (NCT03994796) [39].

### 3.3. Intrathecal Therapy

Intrathecal therapy involves the delivery of therapeutic agents directly into the cerebrospinal fluid/subarachnoid space. This is typically conducted either via lumbar puncture or via a port implanted into the lateral ventricle (i.e., Ommaya port). The rationale for intrathecal therapy is to achieve therapeutic concentration of drugs in the cerebrospinal fluid as there exists a blood cerebrospinal fluid barrier analogous to the blood–brain barrier that impedes CSF entry of therapeutics when delivered systemically. 

In order to be a proper candidate for intrathecal drug delivery, patients must have nonobstructive CSF flow. Additionally, intrathecally administered agents do not penetrate the brain parenchyma beyond 1–2 mm, and therefore currently intrathecal agents should not be considered in the treatment of brain metastases nor bulky LMD. Table 2 outlines a sampling of therapeutics used in the intrathecal setting. 

## 4. Prognosis of Patients with HER2+ Breast Cancer with Metastases to the Brain

Since the advent of Trastuzumab in the late twentieth century, prognosis of HER2+ breast cancer has improved significantly by measure of overall survival (OS), progression-free survival (PFS), and recurrence-free survival [47]. However, prognosis among all patients with breast cancer with metastases to the brain remains poor, with survival of about 3–36 months [48]. Prognosis depends on multiple factors, such as a patient’s age, performance status; overall number of brain metastasis; time from diagnosis to CNS metastasis; evidence of leptomeningeal disease, which is a poor prognostic factor; number of brain metastases; and control of extracranial disease [49]. Interestingly, HER2-positive patients with brain metastasis have better outcomes than patients with hormone-receptor-negative or triple-negative disease [50]. Prognostic models such as the Modified Breast GPA, have been used to confirm risk factors associated with overall survival [51].

Prior to systemic HER2-targeted therapy, OS was likely comparable between patients with brain metastasis who had HER2-positive and HER2-negative disease. This is evidenced by a retrospective analysis by Dawood et al. that demonstrates similar risk of death and overall survival between patients with HER2-negative disease and patients with HER2-positive disease who did not receive first-line targeted therapy [52]. In another study by Bergen et al., patients treated with trastuzumab/pertuzumab had OS of 44 months compared to 17 months with other HER2-targeted therapy and 3 months with no targeted therapy (*p* < 0.001) [53]. Further subgroup analysis revealed that patients with HER2-positive patients with CNS disease who were diagnosed with breast cancer before the year 2000 had an OS of 12 months compared to 22 months for patients diagnosed after the year 2010, likely due to the advent of HER2-targeted therapy (*p* = 0.002).

Finally, prognosis among patients with HER2+ patients with brain metastasis also depends on hormone receptor co-expression. A retrospective analysis of outcomes in 4033 patients from 2008 to 2014 found that HER2-positive metastatic disease was associated with longer median OS compared to HER2-negative disease: HER2+/HR+ disease 18.9 months, HER2+/HR− disease 13.1 months, HER2−/HR+ 7.1 months, and TNBC 4.4 months [54]. Unfortunately, after a median follow-up of 30 months, only 26% of all patients with brain metastasis were living. Further advancements in treatment and HER2-directed therapies will hopefully lead to continued improvements in outcomes for this patient population. 

## 5. Ongoing Studies/Future Directions

As therapeutics have evolved over the past decade, so have the clinical trials that ultimately bring these treatments to patients. There has been a movement to include more CNS metastasis patients in clinical trials, as well as trials specifically dedicated to CNS metastases.

Table 3 herein outlines a sampling of the current clinical trials in the metastatic HER2-positive space that either include CNS metastases in their analysis or are specifically designed to evaluate CNS metastases. These trials, along with furthered basic science and translational research in the field, provide hope for better outcomes for years to come for these HER2-positive CNS metastasis patients. 

## 6. Conclusions

The past twenty years have proven to be revolutionary in the care of HER2-positive CNS metastases, with multiple new approved therapies for metastatic HER2 and select therapeutics with CNS benefit as well. That being said, there is still much work to do with respect to prognosticating, screening, diagnosing and treating these patients. CNS disease in HER2-positive patients remains the primary cause of mortality, and together with better scientific understanding and therapeutic advances, a great deal more progress can be made to improve outcomes in this population.

## Figures and Tables

**Figure 1 cancers-15-02908-f001:**
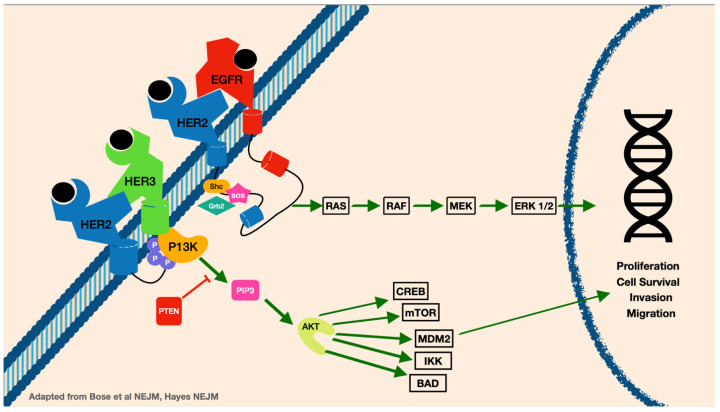
The HER2 signaling pathway.

**Table 1 cancers-15-02908-t001:** HER2-directed therapies for metastatic breast cancer.

Targeted Therapy for HER2-Positive Breast Cancer				
Regimen	Mechanism	Indications	Adverse Effects	Trials Included
Trastuzumab, pertuzumab, and taxane	Combination of MOAbs with two separate extracellular domains of HER2, taxane	First-line therapy	Diarrhea, neutropenia, rash, mucositis, alopecia, neuropathy, LV dysfunction *	APHINITY, CLEOPATRA, PERUSE
Ado-trastuzumab emtansine (T-DM1)	Antibody–drug conjugate with thioether linker and microtubule inhibitor	Second-line therapy in advanced/metastatic disease, adjuvant therapy	Thrombocytopenia, LFT abnormalities, increased bleeding risk independent of PLT count, LV dysfunction *	MARIANNE, TH3RESA, EMILIA, Destiny-Breast03
Trastuzumab deruxtecan (T-DXd)	Antibody–drug conjugate with tetrapeptide-based linker and topoisomerase I inhibitor	Second- or third-line therapy for unresectable/metastatic disease that progressed on prior regimens	Interstitial lung disease; must be discontinued after Grade II pneumonitis	Destiny-Breast03
Tucatinib, Trastuzumab, and Capecitabine	Combination of selective oral TKI, HER2 Moab, and oral Fluorouracil prodrug	Second- or third-line therapy for unresectable/metastatic disease, especially in brain metastases	Palmar–plantar erythrodysesthesia, diarrhea, LFT abnormalities	HER2CLIMB
Margetuximab	Fc-engineered HER2 MOAb	Third-line therapy	Fatigue, GI upset, cough, dyspnea, infusion reaction, palmar–plantar erythrodysesthesia, LV dysfunction	SOPHIA
Lapatinib, trastuzumab	Combination of Oral TKI (EGFR/HER2) and MOAb	Third-line therapy	Diarrhea, palmar–plantar erythrodysesthesia	EMILIA
Neratinib, Capecitabine	Combination of Oral TKI and oral Fluorouracil prodrug	Third-line therapy	Diarrhea, acute liver injury	NALA, ExteNET, SUMMIT

* Trastuzumab is associated with systolic dysfunction; dose-independent and reversible on cessation of therapy.

**Table 2 cancers-15-02908-t002:** Select intrathecal therapy options.

Agent	Description	Half-Life in the CSF	Recommended Schedules of Administration	Recommended Prophylaxis of Adverse Events
Cytarabine	Pyrimidine nucleoside analogue	<1 h	10 mg twice weekly (total 4 weeks), then 10 mg once weekly (total 4 weeks), then 10 mg once monthly	None
Liposomal cytarabine *	Pyrimidine nucleoside analogue	14–21 days	50 mg every 2 weeks (total 8 weeks), then 50 mg once monthly	Oral steroids [40]
Methotrexate	Folate antimetabolite	4.5–8 h	10–15 mg twice weekly (total 4 weeks), then 10–15 mg once weekly (total 4 weeks), then 10–15 mg once monthly	Folinic acid rescue [41]
Topotecan	Topoisomerase 1 inhibitor	1.3 h [42,43]	0.4 mg twice weekly × 4–6 weeks, then weekly × 4, then every other weekly × 4 then monthly	
Thiotepa	Alkylating ethyleneimine compound	3–4 h	10 mg once every other week	Given with methylprednisone 40 mg [44]
Trastuzumab	Monoclonal antibody		80 mg twice weekly or 150 mg weekly	None [45,46]

* Currently not commercially available.

**Table 3 cancers-15-02908-t003:** Ongoing clinical trials for HER2-positive cancer with brain metastasis.

Clinical Trials for HER2-Positive Breast Cancer with Metastasis to the Brain							
NCT Identifier	Title	Sponsor	Phase	N	Status	Trial Drug(s)	Results
NCT05800275	Multicentric Single Arm Phase II Study Evaluating the Efficacy of Association of Tucatinib, Capecitabine and Intra-CSF Trastuzumab in HER2 Amplified Breast Cancer Patients With Leptomeningeal Metastases	UNICANCER	II	n/a	Not yet recruiting	Tucabinib, Capecitabine, Trastuzumab	n/a
NCT04158947	A Randomized Study of HER2+ Breast Cancer Patients With Active Refractory Brain Metastases Treated With Afatinib in Combination With T-DM1 vs. T-DM1 Alone	Zhejiang University School of Medicine, Second Affiliated Hospital (Hangzhou, China)	I	n/a	Recruiting	Afatinib, T-DM1	n/a
NCT03696030	A Phase 1 Cellular Immunotherapy Study of Intraventricularly Administered Autologous HER2-Targeted Chimeric Antigen Receptor (HER2-CAR) T Cells in Patients With Brain and/or Leptomeningeal Metastases From HER2 Positive Cancers	City of Hope Medical Center	I	n/a	Recruiting	Autologous HER2-Targeted CAR T Cells	n/a
NCT04588545	Phase I/II Study of Radiation Therapy Followed by Intrathecal Trastuzumab/Pertuzumab in the Management of HER2+ Breast Leptomeningeal Disease	H. Lee Moffitt Cancer Center and Research Institute	I/II	n/a	Recruiting	Radiation therapy, trastuzumab/pertuzumab	n/a
NCT05041842	Treatment With Tucatinib in Addition to Pertuzumab and Trastuzumab in Patients With HER2-positive Metastatic Breast Cancer After Local Therapy of Isolated Brain Progression	UNICANCER	II	n/a	Recruiting	Tucatinib, pertuzumab and trastuzumab	n/a
NCT05018702	A Prospective, Single-arm, Single-center Phase II Clinical Study of Recombinant Humanized Anti-HER2 Monoclonal Antibody-AS269 Conjugate (ARX788) in the Treatment of HER2-positive Breast Cancer Patients With Brain Metastases	Fudan University (Shanghai, China)	II	n/a	Recruiting	ARX788 (conjugate anti-HER2 MoAb)	n/a
NCT03765983	Phase II Trial of GDC-0084 in Combination With Trastuzumab for Patients With HER2-Positive Breast Cancer Brain Metastases	Dana-Farber Cancer Institute	II	n/a	Recruiting	GDC-0084 (PI3K/mTOR inhibitor), Trastuzumab	n/a
NCT04158947	A Randomized Study of HER2+ Breast Cancer Patients With Active Refractory Brain Metastases Treated With Afatinib in Combination With T-DM1 vs. T-DM1 Alone	Zhejiang University School of Medicine, Second Affiliated Hospital (Hangzhou, China)	II	n/a	Recruiting	Afatinib, T-DM1	n/a
NCT03417544	A Phase II Study of Atezolizumab in Combination With Pertuzumab Plus High-dose Trastuzumab for the Treatment of Central Nervous System Metastases in Patients With Her2-positive Breast Cancer	Dana-Farber Cancer Institute	II	n/a	Recruiting	Atezolizumab, Pertuzumab, Trastuzumab	n/a
NCT04622319	A Phase 3, Multicenter, Randomized, Open-Label, Active-Controlled Study of Trastuzumab Deruxtecan (T-DXd) Versus Trastuzumab Emtansine (T-DM1) in Participants With High-Risk HER2-Positive Primary Breast Cancer Who Have Residual Invasive Disease in Breast or Axillary Lymph Nodes Following Neoadjuvant Therapy (DESTINY-Breast05)	Daiichi Sankyo, Inc. (Tokyo, Japan)	III	n/a	Recruiting	T-DXd, T-DM1	n/a
NCT04739761	An Open-Label, Multinational, Multicenter, Phase 3b/4 Study of Trastuzumab Deruxtecan in Patients With or Without Baseline Brain Metastasis With Previously Treated Advanced/Metastatic HER2-Positive Breast Cancer (DESTINY-Breast12)	AstraZeneca (Cambridge, UK)	III	n/a	Recruiting	Trastuzumab Deruxtecan (T-DXd)	n/a
NCT01921335	Phase I Dose-escalation Trial of ARRY-380 in Combination With Trastuzumab in Participants With Brain Metastases From HER2+ Breast Cancer	Dana-Farber Cancer Institute (Boston, MA, USA)	I	41	Active	ARRY-380 (small molecule inhibitor of HER2), Trastuzumab	n/a
NCT04582968	A Phase Ib/II Pilot Study of Pyrotinib Plus Capecitabine Combined With Brain Radiotherapy in HER2 Positive Breast Cancer Patients With Brain Metastases	Fudan University (Shanghai, China)	Ib/II	39	Active	Pyrotinib, Capecitabine	n/a
NCT03190967	Phase I/II Study of T-DM1 Alone Versus T-DM1 and Metronomic Temozolomide in Secondary Prevention of HER2-Positive Breast Cancer Brain Metastases Following Stereotactic Radiosurgery	National Cancer Institute (NCI) (Bethesda, MD, USA)	I/II	12	Active	T-DM1, TMZ	n/a
NCT01494662	A Phase II Trial of HKI-272 (Neratinib), Neratinib and Capecitabine, and Ado-Trastuzumab Emtansine for Patients with Human Epidermal Growth Factor Receptor 2 (HER2)-Positive Breast Cancer and Brain Metastases	Dana-Farber Cancer Institute	II	140	Active	Neratinib, Capecitabine, Ado-Trastuzumab Emtansine	n/a
NCT04752059	Phase II Study of Trastuzumab-Deruxtecan (T-DX; DS-8201a) in HER2-positive Breast Cancer Patients With Newly Diagnosed or Progressing Brain Metastases	Medical University of Vienna	II	15	Active	Trastuzumab-Deruxtecan (T-DXd)	n/a
NCT03691051	Pyrotinib Plus Capecitabine in Patients With Brain Metastases From HER2-positive Metastatic Breast Cancer: a Single-arm, Open-label, Ahead Study	Henan Cancer Hospital (Zhengzhou, China)	II	78	Active	Pyrotinib, Capecitabine	n/a
NCT00470847	A Phase I Study of Lapatinib in Combination With Radiation Therapy in Patients With Brain Metastases From HER2-Positive Breast Cancer	Dana-Farber Cancer Institute	I	35	Completed	Lapatinib, WBRT, trastuzumab	Did not meet criteria for feasibility due to toxicity (7/27 with dose-limiting toxicity); CNS objective response rate was 79% by volumetric criteria, 46% progression-free at 6 mos.
NCT00614978	Phase 1 Study of the Combination of Lapatinib and Temozolomide for the Treatment of Progressive Brain Disease in HER-2 Positive Breast Cancer (LAPTEM)	Jules Bordet Institute	I	18	Completed	Lapatinib, Temozolomide	Regimen well tolerated (MTD not reached); 10/15 patients assessed had SD but median PFS 2.6 months.
NCT01783756	Phase 1b/2 Single-arm Trial Evaluating the Combination of Lapatinib, Everolimus and Capecitabine for the Treatment of Patients With HER2-positive Metastatic Breast Cancer With CNS Progression After Trastuzumab	Jonsson Comprehensive Cancer Center (UCLA)	I/II	19	Completed	Lapatinib ditosylate, everolimus, capecitabine	Regimen well tolerated (MTD reached), 3/11 (27%) with partial response, 7/11 with stable disease, best CNS objective response rate 28%, median PFS 6.2 mos, median OS 24.2 mos. However, 74% of participants pretrial had received lapatinib, capecitabine or both.
NCT01305941	A Phase II Study Evaluating The Efficacy And Tolerability Of Everolimus (RAD001) In Combination With Trastuzumab And Vinorelbine In The Treatment Of Progressive HER2-Positive Breast Cancer Brain Metastases	UNC Lineberger Comprehensive Cancer Center	II	32	Completed	Everolimus, Trastuzumab, and Vinorelbine (vinca)	Intracranial response rate 4% (1 PR), clinical benefit rate 27% at 6 mos, median time to progression 3.9 mos, OS 12.2 mos.
NCT00967031	A Multicenter Phase II Clinical Trial Assessing the Efficacy of the Combination of Lapatinib and Capecitabine in Patients With Non Pretreated Brain Metastasis From HER2 Positive Breast Cancer (LANDSCAPE)	UNICANCER	II	45	Completed	Lapatinib ditosylate, capecitabine	66% (29/44) with PR, 49% (22/44) with grade 3 or 4 treatment-related adverse events (diarrhea, hand–foot), 4 patients had to discontinue due to toxicity, no toxic deaths
NCT01622868	Phase II Randomized Study of Whole Brain Radiotherapy/Stereotactic Radiosurgery in Combination With Concurrent Lapatinib in Patients With Brain Metastasis From HER2-Positive Breast Cancer—A Collaborative Study of NRG Oncology and KROG	National Cancer Institute (NCI) (Bethesda, MD, USA)	II	143	Completed	Lapatinib	No significant difference in OS between arms of WBRT/SRS and WBRT/SRS with lapatinib; overall progression rate 70.4% in WBRT/SRS vs. 79.4% in WBRT/SRS with lapatinib
NCT04420598	Multicenter, Open-Label, Single-Arm, Multicohort Phase II Clinical Trial of Trastuzumab Deruxtecan(DS-8201a) in Human Epidermal Growth Factor Receptor 2 HER2+ Advanced Breast Cancer With Brain Metastases and/or Leptomeningeal Carcinomatosis	MedSIR (Barcelona, Spain)	II	41	Completed	Trastuzumab deruxtecan	As of 10/2021: 16-week PFS 87.5%, overall intracranial ORR was 46.2%; all responders had partial responses, 2/21 (9.5%) suffered grade I pneumonitis/ILD
NCT01441596	Lux-Breast 3; Randomised Phase II Study of Afatinib Alone or in Combination With Vinorelbine Versus Investigator’s Choice of Treatment in Patients with HER2 Positive Breast Cancer With Progressive Brain Metastases After Trastuzumab and/or Lapatinib Based Therapy	Boehringer Ingelheim (Ingelheim am Rhein, Germany)	II	121	Completed	Afatinib, Vinorelbine (vinca)	Patient benefit with afatinib-containing treatment arms (+/− vinorelbine) not different compared to investigator choice of treatment
NCT00263588	A Phase II Study of Lapatinib for Brain Metastases in Subjects With ErbB2-Positive Breast Cancer Following Trastuzumab-based Systemic Therapy and Cranial Radiotherapy	Novartis Pharmaceuticals (Basel, Switzerland)	II	242	Completed	Lapatinib	34 patients with CNS metastases included, ORR 21% with reported clinical improvement in neurologic symptoms; time to progression 22 weeks
NCT02260531	A Phase II Study of Cabozantinib Alone or in Combination With Trastuzumab in Breast Cancer Patients With Brain Metastases	Dana-Farber Cancer Institute (Boston, MA, USA)	II	36	Completed	Cabozantinib (small molecule TKI) Trastuzumab	Insufficient activity among heavily pretreated BCBM patients (median 3 prior lines, craniotomy, WBRT/STS)
NCT02536339	An Open-Label, Single-Arm, Phase II Study of Pertuzumab With High-Dose Trastuzumab for the Treatment of Central Nervous System Progression Post-Radiotherapy in Patients With HER2-Positive Metastatic Breast Cancer (PATRICIA)	Genentech, Inc. (South San Francisco, CA, USA)	II	40	Completed	Pertuzumab, High-Dose (6 mg/kg weekly) Trastuzumab	CNS ORR 11% with 4 PR, clinical benefit rate 68% at 4 months and 51% at 6 months
NCT00820222	A Randomized, Multicentre, Open-Label, Phase III Study of Lapatinib Plus Capecitabine Versus Trastuzumab Plus Capecitabine in Patients With Anthracycline- or Taxane-Exposed ErbB2-Positive Metastatic (CEREBEL)	Novartis Pharmaceuticals (Basel, Switzerland)	III	540	Completed	Capecitabine, lapatinib, trastuzumab	Inconclusive for primary end point (incidence of CNS mets for patients without baseline CNS disease as first site of relapse), PFS longer with trastuzumab–capecitabine compared with lapatinib–capecitabine (HR 1.30, 95% CI 1.04 to 1.64)

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
