# Peer review of "Modern Management and Diagnostics in HER2+ Breast Cancer with CNS Metastasis"

_cancers, 2023, doi:10.3390/cancers15112908_

Round 1

Reviewer 1 Report

This review summarizes the Modern Management and Diagnostics in HER2+ Breast Cancer with CNS Metastasis. Herein authors, Surabhi W and colleagues review the epidemiology of HER2 positive CNS metastases, risk factor and prognosis. They also discuss available treatment options for these patient’s and shortcomings of each therapeutic modality as well as treatments on the horizon. 

In general, the review is interesting and helpful for clinical understanding. This review is acceptable with the following comments:

1.     For Figure 1. The HER2 Signaling Pathway, please provide the first and last name for the first author and year of the publication clearly, (D. Bose/R Bose et al year---), to avoid any conflict or copyright issues.

2.     Please correct the sentence it is bit confusing. Page # 4, line 124-126- “A randomized control trial by Forsyth et al evaluated patients with brain tumors with no prior seizures who did not undergo surgery to either antiepileptic drug or not and results showed that seizures occurred in 26% of patients treated with antiepileptics to 15% in the non-antiepileptic group.”

3.     Throughout the review, there are repetition of the sentences that could be avoided.

4.     The sentences are too long, and some places hard to understand, and it is confusing, and this makes it less interesting as for example one sentence, “from 140-144-, Advancements in HER2 targeted drugs allow patients options for treatments, but it is also necessary to evaluate what systemic therapies a patient has already had prior treatment with. The sequence of treatment and which combination is appropriate for patients should be an individualized approach ideally decided on by a multidisciplinary team”., please correct accordingly (for Patients, noun or possessive), all corrections would likely improve the readability and impact of the review.

Author Response

  1. For Figure 1. The HER2 Signaling Pathway, please provide the first and last name for the first author and year of the publication clearly, (D. Bose/R Bose et al year---), to avoid any conflict or copyright issues.

This is an original figure not previously used. 

  1. Please correct the sentence it is bit confusing. Page # 4, line 124-126- “A randomized control trial by Forsyth et al evaluated patients with brain tumors with no prior seizures who did not undergo surgery to either antiepileptic drug or not and results showed that seizures occurred in 26% of patients treated with antiepileptics to 15% in the non-antiepileptic group.”

This has been updated. See attached paper 

  1. Throughout the review, there are repetition of the sentences that could be avoided.

Thank you for this input. The paper was reviewed to shorten sentences. 

  1. The sentences are too long, and some places hard to understand, and it is confusing, and this makes it less interesting as for example one sentence, “from 140-144-, Advancements in HER2 targeted drugs allow patients options for treatments, but it is also necessary to evaluate what systemic therapies a patient has already had prior treatment with. The sequence of treatment and which combination is appropriate for patients should be an individualized approach ideally decided on by a multidisciplinary team”., please correct accordingly (for Patients, noun or possessive), all corrections would likely improve the readability and impact of the review.

This was revised. Please see attached paper. 

Reviewer 2 Report

This article focuses on the diagnosis and treatment of brain metastases in HER2-positive breast cancer. This is an interesting topic that can attract more scholars and clinicians to pay attention to this field. However, the overall framework of this article is logically chaotic and fails to highlight the main points.

1.The purpose of this article is to focus on the treatment and future prospects of brain metastases in HER2-positive breast cancer patients. The second part of the article spends a lot of space describing the clinical manifestations and diagnostic conditions of brain metastases, without highlighting the characteristics of HER2-positive breast cancer. To better illustrate the main points, the incidence and prognosis of breast cancer brain metastases, followed by the proportion and prognosis differences of brain metastases in different molecular subtypes should be described briefly. Then the clinical characteristics and diagnostic methods of HER2-positive brain metastasis patients could be introduced.

2.    The third part introduces the treatment of HER2-positive brain metastases, including the local and systemic treatments.Prior to this, the differences between the treatment of HER2-positive brain metastases and non-brain metastases, as well as the current methods and effects of brain metastases, could be discussed.

3.The possible mechanisms of HER2-positive brain metastasis occurrence should be discussed, whether it differs from the mechanisms of brain metastasis of other cancer subtypes, which might suggest future treatment directions.

Author Response

This article focuses on the diagnosis and treatment of brain metastases in HER2-positive breast cancer. This is an interesting topic that can attract more scholars and clinicians to pay attention to this field. However, the overall framework of this article is logically chaotic and fails to highlight the main points.

1.The purpose of this article is to focus on the treatment and future prospects of brain metastases in HER2-positive breast cancer patients. The second part of the article spends a lot of space describing the clinical manifestations and diagnostic conditions of brain metastases, without highlighting the characteristics of HER2-positive breast cancer. To better illustrate the main points, the incidence and prognosis of breast cancer brain metastases, followed by the proportion and prognosis differences of brain metastases in different molecular subtypes should be described briefly. Then the clinical characteristics and diagnostic methods of HER2-positive brain metastasis patients could be introduced.

Unfortunately, due to word restrictions and the specific target focusing on Brain metastases as well as HER2 positive breast cancer, there was not space to talk about other types of breast cancer or focus on breast cancer as a whole. The introduction is already quite long and does not have space to talk about brain mets in relation to other molecular subtypes of breast cancer. That overall scope of work can be quite large and unfortunately would not allow enough word count to discuss HER2 positive CNS mets in detail. 

2.    The third part introduces the treatment of HER2-positive brain metastases, including the local and systemic treatments.Prior to this, the differences between the treatment of HER2-positive brain metastases and non-brain metastases, as well as the current methods and effects of brain metastases, could be discussed.

As mentioned earlier, this paper was solely focusing on HER2 positive breast cancer CNS mets. Discussion of HER2 positive breast cancer as a whole would be too large of a scope for this focused review article. 

3.The possible mechanisms of HER2-positive brain metastasis occurrence should be discussed, whether it differs from the mechanisms of brain metastasis of other cancer subtypes, which might suggest future treatment directions.

The mechanism is discussed in the fourth paragraph in the introduction:

There is still significant uncertainty on how cancer cells are able to proliferate from the breast parenchyma and embed into the CNS forming brain metastases. The metabolic, immune, anatomic, and cellular environment of the brain is unique from other areas of the body making cancer cells that are able to metastasize to the brain very unique and advanced.[10] Cancer cells require genetic evolution in order to be able to infiltrate and adapt to the brain’s microenvironment.[10]After hematologic spread from the breast tissue, cancer cells must be able to cross the blood brain barrier which is a challenging obstacle made of capillary endothelial cells,  a neuroglial membrane, astrocytes, pericytes, and glial podocytes.[11] Prior treatment for breast cancer, such as radiotherapy can also lead to disruption in the blood brain barrier and ability for cancer cells to cross with more ease. Upon reaching the CNS, the breast cancer cells are able to invade tissue in the brain by extravasation and have perivascular growth.[12] In order to survive in the brain, breast cancer metastasis may develop properties that allow them to assume overexpression of proteins to adapt to the brain’s microenvironment.[13] Throughout tumor progression in the brain, there is also a component of angiogenesis which leads to disruption of the blood brain barrier leading to heterogeneity in treatment penetration.[12]